

# Gynodioecy in the common spindle tree (*Euonymus europaeus* L.) involves differences in the asymmetry of corolla shapes between sexually differentiated flowers

Jiri Neustupa

Department of Botany, Faculty of Science, Charles University, Prague, Czech Republic

## ABSTRACT

Gynodioecy is typically associated with a smaller perianth size in purely pistillate flowers than in hermaphrodite flowers. However, it is unclear whether this size differentiation is associated with any differences in flower shape between the two sexual groups. A geometric morphometric analysis of the symmetry of tetrameric corolla shapes was used in the study of *Euonymus europaeus* L., Darwin's classical system of floral sexual differentiation. I investigated whether there are any shape differences between the female and bisexual flowers, with respect to both purely symmetric variation involving coordinated shape changes of the four petals and asymmetry among petals within flowers. The corolla shapes of the female and bisexual flowers and the variability among flowers within each sexual group were very similar in the purely symmetric components of shape variation. However, the female flowers were significantly more asymmetric with respect to both the lateral and transversal asymmetry of their corolla shapes. This is the first study to apply geometric morphometrics in the analysis of morphological patterns in a sexually differentiated gynodioecious plant system. The results showed that subtle shape differences in corolla asymmetry differ between the sexual groups and indicate diverging developmental or selection signals between the sexes.

## INTRODUCTION

Sexual differentiation of flowers is typically accompanied by morphological differentiation of their components, such as the perianth, in multiple lineages of angiosperms (*Barrett & Hough, 2013*; *Dufay et al., 2014*). Gynodioecy, a dimorphic sexual system where female and hermaphrodite flowers are borne on different individuals, occurs in approximately 0.6% of angiosperm species (*Delph, 1996*; *Dufay et al., 2014*). The hermaphrodite flowers of gynodioecious species ensure male function during sexual reproduction. They also contain ovules and produce viable seeds, although their number, size, or viability may be significantly lower than that of female individuals (*Asikainen & Mutikainen, 2003*; *Shykoff et al., 2003*; *Arnan et al., 2014*). The female, purely pistillate flowers may either

Corresponding author
Jiri Neustupa,
neustupa@natur.cuni.cz

possess rudimentary staminodes resulting from aborted stamen development or lack any male reproductive structures (*Bell, 1985*; *Delph, 1996*). Gynodioecy is widely considered one of the evolutionary pathways from hermaphroditism to complete dioecy (*Spigler & Ashman, 2012*; *Dufay et al., 2014*). However, in many extant species, it is a stable reproductive system with characteristic sexual dimorphism of flowers.

*Darwin (1877)*, in his classical treatise on the sexual differentiation of angiosperms, noted that flowers of gynodioecious species typically vary in corolla size, with female flowers being smaller than hermaphrodite flowers. *Delph (1996)* reviewed sexual dimorphism in gynodioecious and gynomonoecious species from 30 angiosperm families and found that the petals of hermaphrodites were on average approximately 1.3 times larger than those of pistillate flowers. In general, four different but not mutually exclusive explanations have been proposed for this corolla size dimorphism in gynodioecious species (*Darwin, 1877*; *Delph, 1996*; *Miller & Venable, 2003*; *Arnan et al., 2014*). The purely mechanistic explanation suggests that hermaphrodite flowers tend to be larger because their perianths must enclose a large volume of reproductive parts comprising both ovaries and anthers, whereas the female flowers lack any fully developed male organs (*Delph, 1996*). A second proposed explanation is that female flower size has decreased as a result of a trade-off allowing them to allocate more resources to fruit and seed production (*Miller & Venable, 2003*). However, the decreased size of female corollas may be largely a developmental consequence of stamen abortion (*Plack, 1957*; *Ruan, Qin & Han, 2005*). This explanation, which was already favoured by *Darwin (1877)*, is supported by the developmental connection between the petals and stamens due to the functions of the B-class genes during flower development of dicotyledonous species, in accordance with the ABCE model (*Chanderbali et al., 2016*). Finally, it has been argued that the fitness of hermaphroditic individuals, which ensure male function in gynodioecious species, is more affected than that of female individuals by the frequency of pollinator visits, leading to selection for increased corolla size (*Müller, 1873*; *Delph, 1996*; *Delph & Ashman, 2006*). On the other hand, female fitness may be maximized with relatively fewer visits, imposing weaker selection on their floral display. This idea has led to the conclusion that "flower is primarily a male organ" (*Bell, 1985*). In support of this explanation, multiple studies have documented the discrimination of pollinators against the smaller female flowers in various gynodioecious taxa (*Eckhart, 1991*; *Delph & Lively, 1992*; *Ashman, 2000*; *Asikainen & Mutikainen, 2005*; *Van Etten & Chang, 2014*).

However, the morphological differentiation of the corolla between female and hermaphrodite flowers might involve not only petal size but also shape characteristics. The allometric relation of shape and size is one of the fundamental patterns of morphological variation in the angiosperms (*Niklas, 1994*). Therefore, it can be expected that consistent differences in corolla size between female and hermaphrodite individuals within populations of gynodioecious species might be accompanied by some kind of shape differentiation (*Kamath, Levin & Miller, 2017*). However, shape variation of complex structures, such as the corolla of actinomorphic flowers, involves conceptually different and separable components of symmetric and asymmetric variability. Symmetric components are typical by coordinated changes in the shape of all the petals constituting each corolla

among different flowers. Thus, this kind of shape variation highlights the differences among flowers, while the petals composing each corolla remain identical. On the other hand, asymmetric variation differentiates the shape of the petals within flowers (*Savriama, 2018*). Does the sexual differentiation of flowers between female and hermaphroditic individuals involve differences in the symmetric shape components of their corollas? In addition, are there any differences in the patterns and degree of corolla asymmetry between the female flowers that abort their stamens and do not produce any pollen and the hermaphroditic flowers that are solely responsible for the male function within populations? In this study, these questions were investigated by a geometric morphometric analysis of symmetry in corolla shape variation of a well-known gynodioecious species.

*Euonymus europaeus* L. (the common spindle) is one of *Darwin*'s (*1877*) classical examples of sexual differentiation in angiosperm flower size. The common spindle belongs to Celastraceae and it is a small tree with a temperate Eurasian distribution area (*Popescu, Caudullo & De Rigo, 2016*). The populations are characterized by a stable gynodioecious reproductive system, with individuals consistently producing either hermaphroditic or purely pistillate flowers with rudimentary staminodes (*Darwin, 1877*; *Webb, 1979*; *Thomas, El-Barghathi & Polwart, 2011*). Female individuals compose approximately 20% of a population but generally exhibit higher seed production than hermaphrodites, which possess protandric flowers with functional stamens and pistils (*Webb, 1979*; *Simmons, 2004*). The corollas of the female flowers are consistently smaller than those of hermaphrodite flowers, but no other morphological differences between the sexual forms have been reported in the literature.

*Euonymus europaeus* is considered a species with truly actinomorphic flowers (*Darwin, 1877*; *Thomas, El-Barghathi & Polwart, 2011*; *Idžojtić, 2019*). The four petals constituting each corolla as well as the four sepals and stamens or staminodes are considered to be identical, and the flower does not exhibit any consistent differentiation across the left–right or adaxial-abaxial axis (*Darwin, 1877*; *Simmons, 2004*; *Popescu, Caudullo & De Rigo, 2016*). From a geometric morphometric perspective of symmetry, the two-dimensional frontal view of the common spindle corolla represents an object with four-fold symmetry arranged according to two perpendicular axes, i.e., a structure with four possible symmetry transformations that in the absence of any asymmetry would leave the object unchanged (*Savriama, Neustupa & Klingenberg, 2010*; *Savriama & Klingenberg, 2011*; *Savriama, 2018*; Fig. S1).

In this study, geometric morphometric techniques were applied for the first time, to the best of my knowledge, to a sexually separated plant model system. I investigated whether corolla shape of gynodioecious *E. europaeus* differs between the female and hermaphroditic flowers with respect to both symmetric variation and within-flower asymmetry. The null hypothesis, evaluated through the geometric morphometric analysis of corolla shape, was that the flowers of the females and hermaphrodites differ only in corolla size and that their overall shape features and the asymmetry among the petals constituting each flower are largely identical.

## MATERIALS AND METHODS

### Sampling

The flowers of the common spindle (*E. europaeus* L.) were sampled within a 2.1 ha area of semi-natural broad-leaved vegetation in Central Bohemia, Czech Republic (50.19583N, 14.03212E), at an altitude of 320–338 m above sea level. Within this study area the sampled trees were at least 10 m from one another. Sampling was performed on May 23–24, 2019, when most individuals at the locality were in peak flowering phase (Fig. 1). In total, 169 flowers from 16 individuals (8 females, 8 hermaphrodites) were sampled. The sexual identity of each individual was determined by inspecting multiple flowers from different cymes in different stages of flower development. The flowers were photographed at anthesis in planar view and at a fixed distance of 20 cm using a Canon EOS 1200D (Canon Inc., Oita, Japan) digital camera with an EFS 18-55 mm objective. To assess imaging error, each flower was photographed twice independently.

### Digitisation of corolla forms

The corolla forms were digitised using the semi-automatic *draw-background-curve* function of TpsDig, version 2.22 (*Rohlf, 2015*). The curves were drawn along the outlines of four petals constituting individual corollas. Then, 50 equidistant points were placed along the outline of each petal (Fig. 2). Thus, the entire configuration, describing the outline form of each analysed corolla, consisted of 200 points (Data S1). To assess digitisation error, all outlines were registered twice, once in a clock-wise direction and once in a counter clock-wise direction. Then, the points from the later digitisation were re-labelled to match the labels of the clockwise digitisation.

### Geometric morphometrics

Geometric morphometric analysis of tetrameric corolla in *E. europaeus* requires four symmetry transformations prior to actual generalized Procrustes analysis (GPA). These transformations correspond to (1) identity, (2) reflections across horizontal and (3) vertical axes of symmetry (Fig. 2), and (4) reflection across both of these axes (*Savriama, Neustupa & Klingenberg, 2010*; *Savriama & Klingenberg, 2011*; *Klingenberg, 2015*; *Neustupa, 2017*; *Savriama, 2018*). Note that the symmetry transformations no. 2–4 have to be accompanied by appropriate re-labelling of individual points delimiting the configurations (*Klingenberg, Barluenga & Meyer, 2002*; *Savriama, Neustupa & Klingenberg, 2010*; *Klingenberg, 2015*; *Savriama, 2018*). Consequently, the resulting dataset consisted of 169 (original flowers) ×2 (imaging events) ×2 (digitisations) ×4 (symmetry transformations) = 2,704 configurations.

GPA of the configurations involved an additional step that allowed the points along the petal outlines to slide iteratively along the outline tangents so that their final position yielded the smoothest possible deformation of the actual configuration from the mean shape of the entire dataset. This position is characterized by the lowest possible bending energy between the mean shape and each configuration (*Bookstein, 1997*; *Perez, Bernal & Gonzales, 2006*; *Gunz & Mitteroecker, 2013*). Two marginal points at the base of each petal were treated as fixed landmarks, and 48 points in between were treated as semilandmarks

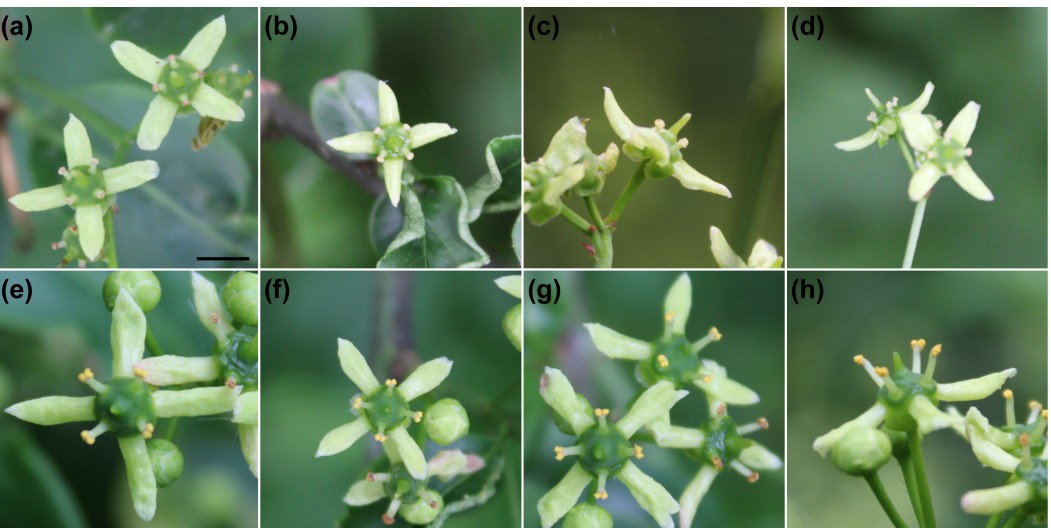

**Figure 1** **Flowers of *Euonymus europaeus*.** (A–D) Female flowers with a central pistil and four pollenless staminodes. (E–H) Bisexual flowers bearing both the central pistil and four stamens. Scale bar (included in A) = three mm.

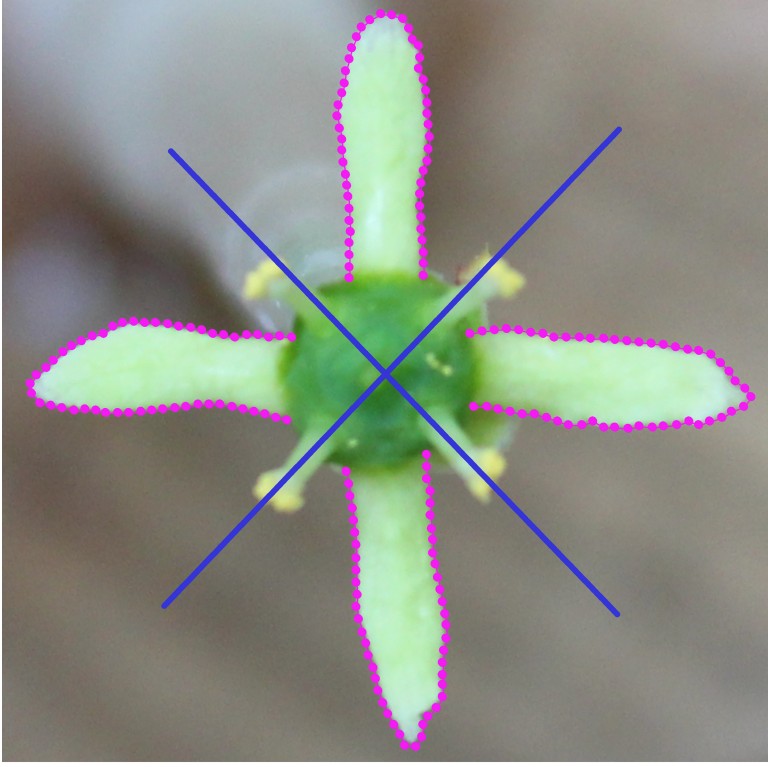

**Figure 2** **A flower of *Euonymus europaeus* with 50 equidistant semilandmarks placed along the outline of each petal.** Two axes of symmetry dissecting the tetrameric corolla are indicated by the blue lines.

(Fig. 2). The sizes of the configurations were measured as centroid size (CS), the square root of the sum of squared distances of each landmark from their centroid (*Zelditch, Swiderski & Sheets, 2012*). GPA with sliding semilandmarks was conducted using the function *gpagen* implemented in the *geomorph* package, version 3.0.7 (*Adams & Otárola-Castillo, 2013*), of R, version 3.5.1 (*R Development Core Team, 2018*).

## Analysis of measurement error and variation among individuals

To analyse the effects of repeated imaging and digitization on shape variation, the configurations resulting from the four symmetry transformations of each object were averaged (*Savriama, 2018*). The shape and size data of these symmetrised configurations were evaluated by two parallel multivariate non-parametric type I Procrustes analysis of variance (ANOVA) models (*Klingenberg, 2015*). The main fixed effect of "sex" distinguished the flowers taken from the female and hermaphroditic individuals. The effect of "tree", nested within the main fixed factor, evaluated the differences among flowers taken from different individuals. The factor "flower" was nested within "tree" and accounted for the differences among individual flowers within trees. Two factors were used to evaluate measurement error: the imaging error originating from repeat photographs of each analysed flower and the digitisation error arising from imprecision during the registration of the corolla outlines (*Berger et al., 2017*; *Savriama, 2018*). Computation of the F-ratios and corresponding *p*-values for individual effects reflected the nested structure of the data, with *p*-values based on the random distribution created by permutations of the nearest subordinate nested factors. These analyses were based on 999 permutations. The function *procD.lm* implemented in the package *geomorph*, version 3.0.7, was used for this analysis.

Differences in corolla shape between the female and hermaphroditic flowers were also evaluated by multivariate regression of the symmetrised shape data onto the single independent binary factor of sexual group membership. The analysis was implemented in TpsRegr, version 1.42 (*Rohlf, 2015*), and significance of the model was assessed by 999 permutations on the Goodall's F ratio (*Zelditch, Swiderski & Sheets, 2012*). Patterns of variation differentiating the sexual morphs were visualized by deformation grids from the overall consensus configuration in TpsRegr.

## Analysis of corolla shape symmetry

Principal component analysis (PCA) of shape configurations representing a complete symmetry group yields principal components (PC) corresponding to individual orthogonal subspaces that uniquely describe different types of symmetry and asymmetry among the repeated parts of the analysed structure (*Mardia, Bookstein & Moreton, 2000*; *Savriama, Neustupa & Klingenberg, 2010*; *Klingenberg, 2015*; *Savriama, 2018*). In case of the common spindle corolla, the repeated parts are represented by four petals symmetrically arranged around the reproductive organs. The completely symmetric subspace involves variation among the flowers, wherein the shapes of all four petals are changed in a coordinated fashion and remain mutually identical. This kind of variation corresponds to an ideal actinomorphy or disymmetry of corolla morphology (*Culbert & Forrest, 2016*). The positions of four reflected and re-labelled copies of a single corolla on these PCs are exactly identical.

The lateral asymmetry of corolla shape is characterized by shape differentiation between two pairs of adjacent petals within flowers. The flowers of *E. europaeus* lack any differentiation between the two axes of symmetry dissecting the corolla into two halves, each consisting of two petals (Fig. 2). Thus, PCs describing these lateral asymmetric patterns cannot be distinguished into those highlighting either the left–right or the adaxial-abaxial asymmetry of corolla shapes. Thus, these two subspaces must be combined into a single pattern representing the overall lateral asymmetry. The third group of PCs represents the transversal asymmetry within flowers. The shape variation in this subspace highlights differentiation between two pairs of opposite petals (*Savriama, 2018*). Two pairs of reflected and re-labelled configurations representing the symmetry group of each analysed corolla on the asymmetric PCs are positioned as mirror images of each other in the opposite parts of the shape space (*Savriama & Klingenberg, 2011*).

The amount of each kind of shape asymmetry in individual objects can be evaluated as the sum of their Euclidean distances on PCs representing a particular subspace of asymmetric variation from the midpoint represented by the ideally symmetric consensus configuration (*Neustupa, 2017*). Relative amounts of individual subspaces can be quantified by summing the variation spanned by PCs belonging to each of these subsets (*Savriama, 2018*). It should be noted that the PCA-based approach quantifies the overall amounts of asymmetry in individual subspaces but does not distinguish between directional and fluctuating asymmetry (*Savriama et al., 2012*).

In this study, the GPA-aligned configurations of the complete symmetry groups of female and hermaphrodite flowers were used. The resulting PCs were classified into three subspaces (symmetry, lateral asymmetry, transverse asymmetry). Differences in the variability of female and hermaphrodite flowers in each of these subspaces were quantified by the test for homogeneity of multivariate dispersions, an analogue of Levene's test for homogeneity of variances (*Oksanen et al., 2019*). In this analysis, the PC axes describing the individual patterns of symmetry or asymmetry were considered separately, and the multivariate dispersion in the scores of female and hermaphrodite flowers on these axes was compared using the function *betadisper* of the package *vegan*, version 2.5-4 (*Oksanen et al., 2019*) in R, version 3.5.1 (*R Development Core Team, 2018*). The *p*-values were based on 9,999 random permutations of the original group assignments. In the analysis of symmetric variation, the null hypothesis was that females and hermaphrodites do not differ in the amount of shape variability among flowers. In the analysis of asymmetric subspaces, the null hypothesis was that the amount of within-flower asymmetry in petal shape does not differ between the two sexual morphs.

The multivariate dispersion was visualized by two-dimensional non-metric multidimensional scaling (NMDS) of the PC scores of corolla shape in each of the three analysed subspaces. This ordination technique iteratively reduces the multivariate structure of an original dataset into two dimensions that approximate their dispersion (*Borg & Groenen, 2005*). The resulting two-dimensional plots were very informative for the comparison of the amount of variation between the sexual groups. These NMDS analyses were conducted in PAST, version 2.17c (*Hammer, Harper & Ryan, 2001*).

**Table 1 Results of analysis of variance evaluating variation in centroid size of the corolla in female and hermaphroditic flowers of *Euonymus europaeus*.**

| Source | Df | SS | MS | $\eta^2$ | F | *p*-value |
|---|---|---|---|---|---|---|
| Sex | 1 | $7.01 \times 10^7$ | $7.01 \times 10^7$ | 0.635 | 36.999 | 0.002 |
| Tree (sex) | 14 | $2.65 \times 10^7$ | $1.89 \times 10^6$ | 0.240 | 21.578 | 0.001 |
| Flower (tree) | 153 | $1.34 \times 10^7$ | $8.78 \times 10^4$ | 0.122 | 39.032 | 0.001 |
| Imaging error | 169 | $3.80 \times 10^5$ | $2.25 \times 10^3$ | 0.003 | 19.107 | 0.001 |
| Digitising error | 338 | $3.98 \times 10^4$ | $1.18 \times 10^2$ | 0.0004 | | |
| Total | 675 | $1.10 \times 10^8$ | | | | |

Notes.
Df, degrees of freedom; SS, sum of squares; MS, mean squares; $\eta^2$, percentage of variance explained by individual effects.

**Table 2 Results of Procrustes analysis of variance of symmetric corolla shape variation in female and hermaphroditic flowers of *Euonymus europaeus*.**

| Source | Df | SS | MS | $\eta^2$ | F | *p*-value |
|---|---|---|---|---|---|---|
| Sex | 1 | 0.054 | 0.054 | 0.024 | 1.384 | 0.225 |
| Tree (sex) | 14 | 0.542 | 0.039 | 0.243 | 3.699 | 0.001 |
| Flower (tree) | 153 | 1.602 | 0.010 | 0.717 | 103.683 | 0.001 |
| Imaging error | 169 | 0.017 | 0.0001 | 0.008 | 1.858 | 0.001 |
| Digitising error | 338 | 0.018 | 0.00005 | 0.008 | | |
| Total | 675 | 2.234 | | | | |

Notes.
Df, degrees of freedom; SS, sum of squares; MS, mean squares; $\eta^2$, percentage of variance explained by individual effects.

## RESULTS

The corolla size of the female flowers was approximately 15.5% smaller than that of the hermaphroditic flowers. This difference was highly significant in the nested ANOVA that evaluated variation in CS of the configurations between two sexual groups against the differences among individual trees (Table 1). Furthermore, within each sexual group, individuals differed in flower size. The imaging and digitising errors proved to be relatively negligible with regard to size variation among flowers (Table 1). The symmetric corolla shapes of the two sexual morphs were very similar, as their observed difference was not significant compared to the variation between individual trees (Table 2). However, the trees within each group differed in their mean corolla shapes, and the variation among flowers was considerably greater than the error arising from the repeated imaging and digitisation of corolla forms (Table 2).

Multivariate regression of symmetrised corolla shapes against the sexual groups revealed subtle but significant differences (Goodall's $F = 4.21$, $p = 0.005$) that accounted for 2.45% of the total variation explained by this regression model. In particular, the petals of female flowers had slightly thicker basal parts than those of hermaphroditic flowers (Fig. 3).

The purely symmetric variation, which maintained all four petals within each corolla identical, comprised 45.19% of total variability in the shape space. In this subspace, all four configurations forming the symmetry group of each corolla had the same scores on individual PCs (Fig. 4A). The single most prominent shape pattern within this subspace,

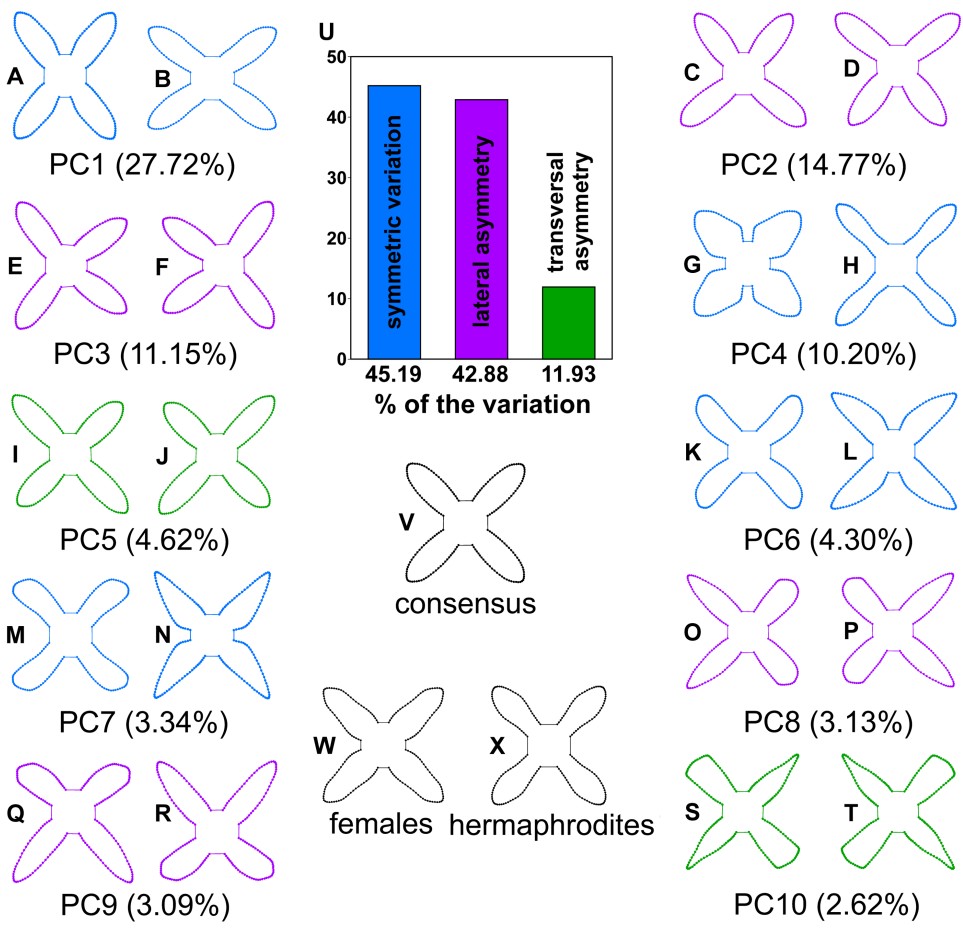

**Figure 3 A scheme showing the configurations of *Euonymus europaeus* corolla reconstructed by the principal component analysis of biradial symmetry and by the linear discriminant analysis of two sexual morphs.** (A–T) The first ten principal components are shown. The configurations illustrate shapes typical for the most marginal occupied positions within the shape space. Blue configurations indicate the components depicting the purely symmetric variation. Violet indicates the components spanning two subspaces of lateral asymmetry. Green depicts the components showing the patterns of transversal asymmetry in corolla shapes. (U) The histograms showing relative amounts of variation spanned by symmetry and two types of within-flower asymmetry. (V) Configuration of the consensus corolla shape and the typical features differentiating (W) the female and (X) hermaphroditic flowers in the linear discriminant analysis. The differences between the sexual morphs are four times enhanced for better visibility.

illustrated by PC1, described changes in corolla shape resulting from different orientation of flowers that deviated from an actinomorphic shape depicted by the consensus configuration to a disymmetric corolla morphology (Fig. 3). The three subsequent PCs belonging to this subspace, namely, PC4, PC6 and PC7, described shape trends involving coordinated changes in the shape of individual petals. Notably, PC4 illustrated variation between flowers, differentiating those with thick, short and oval petals from those with elongate, thin petals (Fig. 3). PC6 and PC7 exhibited very similar trends corresponding to changes among slightly elongate, obtuse petals and those with pointed, bilaterally asymmetric shapes (Fig. 3).

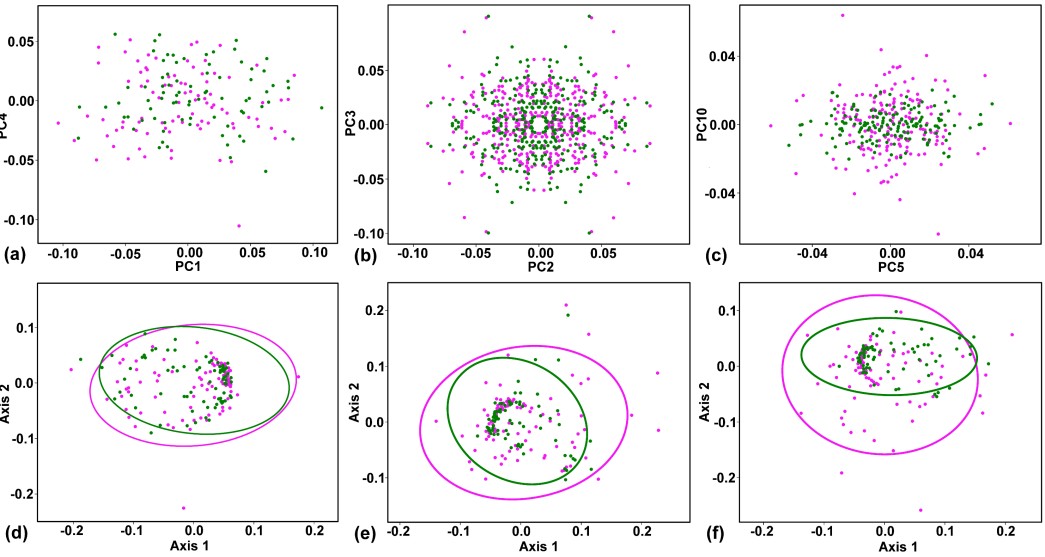

**Figure 4** **Plots showing the ordination structure and the amounts of variability in corolla shapes of sexually differentiated flowers of *Euonymus europaeus*.** Positions of the purely pistillate flowers (magenta) and hermaphroditic flowers (green) are depicted. (A) Two principal components (PCs) describing the symmetric variation show a pattern with each data point representing four symmetry transformations of a single corolla. (B) PCs describing two mutually orthogonal patterns of lateral asymmetry. Each corolla is represented by four data points occupying mirror positions on each axis. (C) PCs representing the laterally asymmetric subspace of the variation. Four symmetry transformations of each corolla occupy two data points in mirror positions in the shape space. (D) Two-dimensional non-metric multidimensional scaling (NMDS) plot showing the overall amounts of symmetric variation in female (red) and hermaphroditic (green) flowers. Kruskal's stress = 0.22. (E) NMDS plot showing the overall amounts of variation in PCs spanning the subspaces of lateral asymmetry in both sexual groups. Kruskal's stress = 0.24. (F) NMDS plot showing the overall amounts of variation in PCs spanning the transversal asymmetry of corolla shapes in both sexual groups. Kruskal's stress = 0.26. The 95% variation ellipses for each sexual group are depicted in the NMDS plots.

Together, the two subspaces of lateral asymmetry described 42.88% of the total variation in corolla shape. The laterally asymmetric axes were delimited in pairs differing by their orientation across two ambiguous axes of symmetry. PC2 and PC3 described changes to corolla shapes with laterally compressed adjacent petals (Fig. 3). Each data point on any of the asymmetric PC axes corresponded to two copies of a particular object. The respective reflection of that object is placed in a mirror position. As the lateral asymmetric axes examined in this study spanned two orthogonal subspaces depicting these asymmetric patterns, the ordination plot of PC2 vs. PC3 shows each object in four symmetrically arranged points corresponding to their mirrored positions on each axis (Fig. 4B). Similarly, PC8 and PC9 illustrated the pattern differentiating between two adjacent short and obtuse petals and the opposite pair exhibiting elongated and pointed shapes (Fig. 3). Transversal asymmetry accounted for 11.93% of the total variation in corolla shape. The two most important shape trends in this subspace were represented by PC5 and PC10 (Fig. 3). PC5 described within-flower asymmetry characterized by the elongation of two opposite petals relative to the other, transversally arranged pair. Similarly, PC10 illustrated asymmetry

**Table 3  Results of tests for homogeneity of multivariate dispersions.** The tests evaluated the amounts of corolla shape variation in different subspaces of morphometric symmetry and asymmetry between the purely pistillate and hermaphroditic flowers of *Euonymus europaeus*.

| | Average ED to centroid | | Diff. in means | 95% C.I. of diff. in means | | P-value |
|---|---|---|---|---|---|---|
| | **FM** | **HM** | | **LR** | **UR** | |
| Symmetric variation among flowers | 0.0561 | 0.0511 | 0.0050 | −0.0016 | 0.0116 | 0.1323 |
| Lateral asymmetry within flowers | 0.0580 | 0.0467 | 0.0113 | 0.0053 | 0.0173 | 0.0010 |
| Transversal asymmetry within flowers | 0.0301 | 0.0244 | 0.0058 | 0.0024 | 0.0091 | 0.0006 |

**Notes.**
C.I., confidence interval; FM, females; HR, hermaphrodites; LR, lower range; UR, upper range.

between the pair of transversally arranged petals with pointed shapes and their counterparts with obtusely widened morphology. The ordination plot of these two PCs belonging to the same asymmetric subspace shows a transversally symmetric arrangement of points, each corresponding to two copies of individual objects mirrored by their corresponding reflections on each axis (Fig. 4C).

The tests of the homogeneity of multivariate dispersions showed contrasting patterns between the subspaces of symmetric and asymmetric variations. In the totally symmetric subspace, there was only a slight difference in the amount of variation between the female and hermaphroditic flowers (Table 3, Fig. 4D). Compared with the hermaphrodite flowers, the female flowers were approximately 9.8% more variable among each other. However, the confidence intervals for this difference included zero, and the permutation test yielded a *p*-value for this difference of well above the 0.05 threshold. Thus, the null hypothesis that the dispersion of the female and hermaphrodite symmetric configurations was equal could not be rejected. In contrast, female corolla shapes were considerably more asymmetric than were hermaphrodite corolla shapes in both asymmetric patterns (Table 3, Figs. 4E–4F). The multivariate dispersion of female corolla shapes was approximately 23.4% and 24.2% higher than that of hermaphrodite corolla shapes in transversal and lateral asymmetry, respectively. In both these subspaces, the difference was highly significant (Table 3). Consequently, the null hypothesis of equal mean within-flower asymmetry of corolla shapes between the two sexual groups was rejected.

## DISCUSSION

The attributes of the flowers of the two sexes in the studied population corresponded to *Darwin*'s (*1877*, p. 288, Fig. 12) original description and subsequent studies (*Webb, 1979*; *Thomas, El-Barghathi & Polwart, 2011*). The data presented in this study confirmed consistent differences in corolla size between the female and hermaphroditic flowers of the common spindle tree, which have been reported in different regions of this plant's distribution (*Darwin, 1877*; *Webb, 1979*; *Lloyd, Webb & Primack, 1980*; *Thomas, El-Barghathi & Polwart, 2011*; *Popescu, Caudullo & De Rigo, 2016*). Furthermore, it was also confirmed that *E. europaeus* in the studied area consistently exhibits gynodioecious reproductive features, with purely pistillate flowers consistently being developed on different individuals than those developing bisexual flowers.
Interestingly, the marked size difference between the female and hermaphroditic flowers was not accompanied by any pronounced diversification between morphs in average corolla shape. In fact, the symmetric corolla shapes of females were very similar to those of hermaphrodites, with the sexual group explaining only approximately 2.45% of the variation in shape in the multivariate regression model. Furthermore, the amount of symmetric shape variability among flowers was not significantly different between the sexual groups. These results suggest that the species-specific average corolla shapes of *E. europaeus* are largely consistent regardless of sexual identity of flowers.

However, the analyses of within-flower asymmetry among the four petals constituting the tetrameric corolla of the common spindle revealed significant differences. In both mutually orthogonal subspaces of asymmetric variation, the females consistently had more asymmetric corolla shapes than did the bisexual individuals. Interestingly, the two subspaces of lateral asymmetry represented a considerably greater part of the total variability than the twice the amount of the variation spanned by the subspace of transversal asymmetry. Given the ambiguity of the two axes of symmetry dissecting the tetrameric flowers of *E. europaeus*, the components of lateral asymmetry simply describe the shape patterns in which the adjacent petals are more similar than the opposite pair. One possible underlying cause of increased lateral asymmetry may be cryptic and very subtle quantitative zygomorphy of corolla shapes. Such zygomorphy, if present, could not have been detected by the visual inspection of individual flowers, but it may be reflected by increased variation spanned by the components of lateral asymmetry. However, a more plausible explanation of the higher representation of lateral asymmetry involves the effects of environmental heterogeneity around individual flowers. *Tucić et al. (2018)* recently showed that solar irradiance, which on the northern hemisphere is higher on the southern side of flowers, caused consistent asymmetry of flower parts in *Iris pumila*. A similar mechanism might operate in the actinomorphic corolla of the common spindle, leading to the increased within-flower asymmetry of petals related to their orientation. Even in an ideally actinomorphic corolla *bauplan*, this phenomenon can be expected to result in an increased proportion of lateral asymmetry relative to transverse asymmetry. If this explanation holds, it can be predicted that future studies of symmetry in tetrameric corolla shapes in actinomorphic plants in non-tropical latitudes will typically detect a higher proportion of lateral asymmetric variation than of transverse asymmetric variation.

However, the higher asymmetry of female flowers in both subspaces is unlikely to be attributable to local environmental heterogeneity. The individual trees in this study were growing in the same habitat and did not differ in overall habit, morphology, or cyme position. In addition, any localised environmental heterogeneity, such as differences in irradiation between different corolla parts, would have hardly influenced the components of the transversal asymmetry. However, this subspace was typical by the elevated asymmetry levels of the female flowers, just like the components of the lateral asymmetry. Therefore, it is unlikely that local environmental effects were responsible for the differences between flowers with differing sexual identities. Thus, it is likely that the observed difference was a result of the gynodioecious reproductive strategy of the studied model system. The developmental explanation of size differences between female and hermaphroditic

flowers might be applicable in this case. If the replacement of stamens by staminodes and subsequent lack of pollen production in purely pistillate flowers is associated with the development of petals via the function of the B-class genes (*Chanderbali et al., 2016*) and thus responsible for their smaller size (*Delph, Touzet & Bailey, 2007*), it might also lead to decreased morphological integration of the corolla, resulting in higher shape asymmetry. Similar patterns were recently observed in bisexual *Geranium robertianum* L., where the amount of corolla shape asymmetry was higher in smaller flowers with relatively lower pollen production (*Frey & Bukoski, 2014*).

Furthermore, the smaller size and greater asymmetry of purely pistillate corollas, related to the changes in the function of the B-class genes, might also lead to the lower visual attractiveness of the female flowers to insect pollinators. Relevant data are not available in *E. europaeus*, but in other gynodioecious species, such as *Fragaria virginiana* Mill., *Geranium sylvaticum* L., and *G. maculatum* L., discrimination of pollinators against smaller female flowers has been reported (*Ashman, 2000*; *Asikainen & Mutikainen, 2005*; *Van Etten & Chang, 2014*). In addition, it has been repeatedly observed that the foraging behaviour of insect pollinators may reflect intraspecific variability in tetrameric corolla shapes (*Gómez, Perfectti & Camacho, 2006*; *Gómez et al., 2008*). Pollinators are known to favour flower models with lower asymmetry, even in the absence of any rewards, such as increased nectar production (*Møller & Eriksson, 1995*; *Møller & Sorci, 1998*). In the case of gynodioecious *E. europaeus*, this phenomenon would mean that higher selection pressure for frequent visitation by pollinators maintains corolla symmetry at higher levels in hermaphroditic flowers than in female ones.

The observed patterns did not support the purely mechanistic explanation of morphological differentiation between the sexual groups (*Delph, 1996*). If the corollas of female flowers are generally smaller than the corollas of the hermaphrodites solely because the flower volume occupied by the sexual organs is smaller in females, there is no reason to expect any increase in shape asymmetry among the petals of purely pistillate flowers relative to that of the hermaphrodites.

In addition to stable gynodioecy, transitional patterns of sexual differentiation have been reported in other taxa with actinomorphic corolla morphology. It might be interesting to study these systems using similar analytical procedures of geometric morphometric decomposition of shape symmetry (*Klingenberg, 2015*; *Savriama, 2018*; *Tucić et al., 2018*). For example, a transition from fully bisexual to purely pistillate flowers within a gradient of individuals differing in the number of active stamens has been identified in populations of *Geranium sylvaticum*, which typically exhibits a pentameric corolla morphology (*Asikainen & Mutikainen, 2003*; *Asikainen & Mutikainen, 2005*). In addition, several co-existing sexual types, including purely unisexual, inconstant unisexual, ambisexual and bisexual individuals, have been reported in *Thymelaea hirsuta*, a species with tetrameric actinomorphic flowers (*Shaltout & El-Keblawy, 1992*; *Dommée et al., 1995*). In these systems, the possible effects of stamen abortion on within-flower corolla asymmetry could be tested in complex settings, which might confirm or disprove the generality of this phenomenon and its developmental explanation.

## CONCLUSIONS

This pioneering study illustrated that geometric morphometric analyses of corolla shape symmetry in taxa with sexual differentiation may yield interesting and novel insights into phenotype evolution of unisexual and combined reproductive systems in angiosperms. It has been showed that morphological differentiation of flowers in gynodioecious *E. europaeus* is not limited to size diminution of female flowers but also involves significant differences between sexual types in corolla shape variation. Whereas the symmetric corolla shapes were very similar between the purely pistillate and bisexual flowers, considerable differences in within-flower asymmetry were detected between the sexual forms. The corolla shapes of females were consistently more asymmetric in the subspaces of both lateral and transversal asymmetry. These patterns of a less integrated corolla in purely pistillate flowers suggest the effects of developmental processes related to stamen abortion, which decrease the ontogenetic precision in petal development, potentially acting together with the preference of insect pollinators for symmetric corolla shapes.

The gynodioecy pathway to the fully separated monoecious sexual reproduction of angiosperms represents a unique microevolutionary stage in which flower morphology may gradually diverge due to diverging selection pressures related to the frequency of pollinator visits (*Ashman, 2000*; *Van Etten & Chang, 2014*) and to developmental interactions among flower parts (*Thompson, Rolland & Prugnolle, 2002*; *Ruan, Qin & Han, 2005*). Geometric morphometrics of flower shape symmetry now offers brand new possibilities for the study of these microevolutionary processes.

## ACKNOWLEDGEMENTS

The author thanks Wiley Editing Services for English language editing and style corrections.

### Funding

This work was financially supported by the institutional project Progres Q43 at Charles University, Prague. The funders had no role in study design, data collection and analysis, decision to publish, or preparation of the manuscript.

### Grant Disclosures

The following grant information was disclosed by the author:
Institutional project Progres Q43 at Charles University, Prague.

### Competing Interests

The author declares there are no competing interests.

### Author Contributions

- Jiri Neustupa conceived and designed the experiments, performed the experiments, analyzed the data, prepared figures and/or tables, authored or reviewed drafts of the paper, and approved the final draft.

## Data Availability

The original data is available as a Supplemental File.

## Supplemental Information

Supplemental information for this article can be found online at http://dx.doi.org/10.7717/peerj.8571#supplemental-information.

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
