# Peer review of "Gynodioecy in the common spindle tree (Euonymus europaeus L.) involves differences in the asymmetry of corolla shapes between sexually differentiated flowers"

_PeerJ, doi:10.7717/peerj.8571_

## Round 0.1 · original submission · Minor Revisions

Dear Dr. Neustupa,

I have received two positive reviews for your manuscript. Both suggest minor revisions. I agree. Please add more general information to symmetry definitions like suggested by reviewer one and discuss the point on the impact on pollination made by reviewer two.

Best wishes
Mike Thiv

Reviewer 1 ·

Basic reporting

This manuscript entitled « Gynodiecy in the common spindle tree (Euonymus europaeus L.) involves differences in the asymmetry of corolla shapes between sexually differentiated flowers » by J. Neustupa submitted to PeeRJ is an exemplary interdisciplinary study integrating botany and mathematics.

Figures are relevant, of high quality, well labelled and described.
Figure 1: I could not see the scale bar on the images.

The English language is clear and professional throughout the manuscript, and it was a pleasure to read this article.

Experimental design

The geometric morphometric methods applied here were published recently, and the conclusions that are drawn are convincing, interesting, and open the way to a whole set of studies of floral symmetry using this novel approach.

The research question is well defined, and the experimental design is perfectly adapted to answer the question.

Validity of the findings

The findings are well supported by the data; they are original and well interpreted.

Additional comments

The species that is investigated is an emblematic species in evolutionary biology and belongs to the set of plant taxa which has been studied by Darwin. This will draw the attention of many readers, may they be botanists interested in the evolution of flower shapes, morphometricians keen on investigating organismal symmetry, or ecologists curious of the relations between flower morphology and insect behavior.

The concepts of symmetry handled in this manuscript are different with those that botanists are familiar with, and a bit more complex I have to acknowledge. Many references are made to the works of Yoland Savriama, and symmetry is, in this manuscript, conceived as a concept in crystallography. I would find useful for the readers to include, perhaps as an appendix, an additional figure explaining clearly the symmetric variation, the lateral and transverse asymmetry, etc.

Other comments and suggestions:
l. 55: replace “hermaphrodite […] population” with “hermaphrodite flowers are borne on different individuals”.
l. 108: add the information that the species belongs to family Celastraceae.
l. 118: write Euonymus in full.
l. 123: C4 notation: perhaps refer to the Schoenflies notation and add a citation. C4 usually means something totally different for botanists.
l. 141: how distant were the 16 specimens from one another?
l. 210: “all four petals change their shapes”: I feel the passive voice would better fit here.
l. 279: change “between” to “among”.
l. 295: insert “the” before “elongation”.
l. 369: the stamens are not strictly speaking “aborted”.
l. 429: change “intriguing” to “brand new”, for instance.

Are there any studies of floral development of Euonymus based on which you could compare the size of the meristem for each type of flower (female and hermaphrodite)? This could be another factor influencing the difference in corolla size between floral types.

·

Basic reporting

The paper is well structured and well written. The cotext related to corolla dimorphism of gynodioecious species is well explained and refereed by the relevant citation. However, I find some ambiguity in the explanation of geometric method. For example, in line 100-102 the definition of asymmetric and symmetric components is a bit unclear. The same is true for the line 103-105 where the author addresses the possible association between corolla shape variation and stamen reduction. These parts should be rewritten in a more clear way.

Experimental design

The research question is well-defined and the approach to answering the question is clearly explained. Methods are discussed with sufficient details.
However, in line 161, the transformation method 2 is lacking.

Validity of the findings

The conclusions are linked to the original research questions and often supported by underlying data or relevant literature.
In line 377-379, the author mentions that “the smaller size and greater asymmetry of purely pistillate corollas might be related to the lower selection pressure on female flowers for visual attractiveness to insect pollinators.” For me, the opposite seems more plausible. The smaller size and greater asymmetry which is most likely a result of change in B genes function lead to lower attractiveness for insects.

Additional comments

The author conducted a comparative geometric study of corolla shape between hermaphrodite and female flowers of a gynodioecious species. The study is well-designed, structured and written. My main comments are mentioned in the other sections. I recommend the acceptance of paper after minor revisions.

---

## Round 0.2 · accepted · Accept

Dear Dr. Neustupa,

I am happy with the changes you made as suggested by the reviewers. I just found 3 minor points which should be quickly changed while in production:.

l. 31, 32 delete 2x 'the': Gynodioecy is typically associated with a smaller perianth size in purely pistillate flowers than in hermaphrodite flowers.

l. 369 would have hardly influenced the components ...

Add author names to all other latin plant names like Geranium robertianum, Fragaria virginiana ... Thymelaea hirsuta etc.

Best wishes
Mike Thiv